# Role of mobile genetic elements in the global dissemination of the carbapenem resistance gene *bla*$_{NDM}$

Mislav Acman [1✉], Ruobing Wang[2], Lucy van Dorp [1], Liam P. Shaw [3], Qi Wang[2], Nina Luhmann [4], Yuyao Yin[2], Shijun Sun[2], Hongbin Chen[2], Hui Wang[2] & Francois Balloux [1]

The mobile resistance gene *bla*$_{NDM}$ encodes the NDM enzyme which hydrolyses carbapenems, a class of antibiotics used to treat some of the most severe bacterial infections. The *bla*$_{NDM}$ gene is globally distributed across a variety of Gram-negative bacteria on multiple plasmids, typically located within highly recombining and transposon-rich genomic regions, which leads to the dynamics underlying the global dissemination of *bla*$_{NDM}$ to remain poorly resolved. Here, we compile a dataset of over 6000 bacterial genomes harbouring the *bla*$_{NDM}$ gene, including 104 newly generated PacBio hybrid assemblies from clinical and livestock-associated isolates across China. We develop a computational approach to track structural variants surrounding *bla*$_{NDM}$, which allows us to identify prevalent genomic contexts, mobile genetic elements, and likely events in the gene's global spread. We estimate that *bla*$_{NDM}$ emerged on a Tn*125* transposon before 1985, but only reached global prevalence around a decade after its first recorded observation in 2005. The Tn125 transposon seems to have played an important role in early plasmid-mediated jumps of *bla*$_{NDM}$, but was overtaken in recent years by other elements including IS26-flanked pseudo-composite transposons and Tn3000. We found a strong association between *bla*$_{NDM}$-carrying plasmid backbones and the sampling location of isolates. This observation suggests that the global dissemination of the *bla*$_{NDM}$ gene was primarily driven by successive between-plasmid transposon jumps, with far more restricted subsequent plasmid exchange, possibly due to adaptation of plasmids to their specific bacterial hosts.

[1] UCL Genetics Institute, University College London, Gower Street, London WC1E 6BT, UK. [2] Department of Clinical Laboratory, Peking University People's Hospital, Beijing 100044, China. [3] Department of Zoology, University of Oxford, Oxford OX1 3SZ, UK. [4] Warwick Medical School, University of Warwick, Coventry CV4 7AL, UK. ✉email: mislav.acman.17@ucl.ac.uk

Antimicrobial resistance (AMR) poses a major challenge to human and veterinary health. AMR can be conferred by vertically inherited point mutations or via the acquisition of horizontally transmitted 'accessory' genes, often located in transposons and plasmids. The $bla_{NDM}$ gene represents a typical example of a mobile AMR gene[1]. $bla_{NDM}$ encodes a metallo-β-lactamase capable of hydrolyzing most β-lactam antibiotics. These antibiotics are used as a first-line treatment for severe infections and to treat multidrug-resistant Gram-negative bacterial infections. As such, the global prevalence of bacteria carrying $bla_{NDM}$ represents a major public health concern.

$bla_{NDM}$ was first described in 2008 from a *Klebsiella pneumoniae* isolated from a urinary tract infection in a Swedish patient returning from New Delhi, India[2]. While $bla_{NDM}$ now has a worldwide distribution, most of the earliest cases have been linked to the Indian subcontinent, leading to this region being suggested as the likely location for the initial mobilization event[1,3–6]. NDM-positive *Acinetobacter baumannii* isolates have been retrospectively identified from an Indian hospital in 2005[7], which remain the earliest observations to date. Nevertheless, an NDM-positive *A. pittii* isolate was also collected in 2006 from a Turkish patient with no history of travel outside Turkey[8].

Although no complete genome sequences are publicly available from these earliest observations, the first NDM-positive isolates from 2005 were shown to carry $bla_{NDM}$ on multiple non-conjugative, but potentially mobilizable plasmid backbones[7]. In addition, $bla_{NDM}$ in these early isolates was positioned within a complete Tn*125* transposon with IS*26* insertion sequences (ISs) as well as ISCR*27* (IS-containing common region 27), suggesting the possibility of complex patterns of mobility since the gene's initial integration. Subsequent NDM-positive isolates across multiple species consistently harbour either a complete or fragmented IS*Aba125* (an IS constituting Tn*125*), immediately upstream of $bla_{NDM}$, which provides a promoter region[1,5,9,10]. The presence of IS*Aba125* in some form in all NDM-positive isolates to date and the early observations in *A. baumannii* have led to Tn*125* being proposed as the ancestral transposon responsible for the mobilization of $bla_{NDM}$, and *A. baumannii* as the ancestral host[10,11].

The NDM enzyme itself is of possible chimeric origin[10,11], with the first six amino acids in NDM matching to those in *aphA6*, a gene providing aminoglycoside resistance and also flanked by IS*Aba125*. It is hypothesized that ISCR*27*, which uses a rolling-circle (RC) transposition mechanism[12,13], initially mobilized a progenitor of $bla_{NDM}$ in *Xanthomonas sp.* and placed it downstream of IS*Aba125*[10,11,14,15]. At least 29 distinct sequence variants of the NDM enzyme have been described to date[1,16]. The most prevalent of these variants is the first to have been characterized, denoted NDM-1[17]. Different NDM variants are mostly distinguished by a single amino-acid substitution, apart from NDM-18 that carries a tandem repeat of five amino acids. None of the observed substitutions occur in the active site and their functional impact remains under debate[1].

$bla_{NDM}$ is found in at least 11 bacterial families and NDM-positive isolates have heterogeneous clonal backgrounds, supporting multiple independent acquisitions of $bla_{NDM}$[1]. Although $bla_{NDM}$ has been observed on bacterial chromosomes[18,19] it is most commonly found on plasmids, comprising multiple different backbones or types. Thus far, $bla_{NDM}$ has been associated with at least 20 different plasmid types, predominantly IncFIB, IncFII, IncA/C (IncC), IncX3, IncH, and IncL/M, and also in untyped plasmids[1,4,20–23]. Furthermore, even within the same plasmid type, $bla_{NDM}$ can be found in a variety of genomic contexts, often interspersed by multiple ISs and composite transposons[1,11]. The immediate environment of $bla_{NDM}$ has been reported to vary even in isolates from the same patient[22]. Many mobile elements are thought to play important roles in

dissemination, including IS*Aba125*, IS*3000*, IS*26*, IS*5*, ISCR1, Tn3, Tn*125*, and Tn*3000*[1,14,22,24–26]. It is therefore clear that the spread of $bla_{NDM}$ was, and is, a multi-layer process involving multiple mobile genetic elements—'the mobilome'. $bla_{NDM}$ mobility involves diverse processes, including genetic recombination[27,28], transposition, conjugation and transformation of plasmids[29], transduction[30], and transfer through outer-membrane vesicles[31,32].

Previous surveys of $bla_{NDM}$-positive genomes have led to a better understanding of its evolution[1]. However, a major difficulty, as for other AMR genes, is relating the diverse genomic contexts to temporal evolution. Here, we outline an alignment-based method to identify flanking structural variants and use it to build a history of the insertion and mobilization events. We compile a global dataset of more than 6000 NDM-positive isolates. In line with previous studies, we identify Tn*125*, IS*26* and Tn*3000* as the main contributors to $bla_{NDM}$ mobility but go further and estimate the timing of the initial emergence of $bla_{NDM}$ to pre-1990, around two decades prior to its first detection and rapid dissemination. Our findings suggest that this global spread was driven primarily by transposons, with plasmids playing more of a role in local transmission.

## Results

**A global dataset of $bla_{NDM}$ carriers.** We compiled a dataset of 6155 bacterial genomes (7148 contigs) carrying at least one copy of $bla_{NDM}$ (Fig. 1). These include: published assemblies from NCBI RefSeq[33] ($n = 2632$), NCBI GenBank[34] ($n = 1158$) and Enterobase[35] ($n = 1379$); bacterial genomes assembled using short-read de novo assembly from NCBI's Sequence Read Archive (SRA) ($n = 882$); and newly generated bacterial genomes isolated from 79 hospitalized patients across China and 25 livestock farms assembled using hybrid PacBio-Illumina de novo assembly ($n = 104$) (Supplementary Table 1 and Supplementary Fig. 1). While public genomes have inherent sampling biases, taking advantage of them offers the most comprehensive approach available[1]. Data were included from 251 different Bioprojects, with more than half the samples linked to two large-scale database refinement efforts[36,37]. Quality assessment of $bla_{NDM}$-positive contigs obtained from SRA de novo assemblies showed good contig mapping coverage and did not reveal any problems that could potentially compromise downstream analyses (see Methods and Supplementary Fig. 2).

The dataset included $bla_{NDM}$-positive isolates from 88 states (Fig. 1a) mostly collected in Asia, particularly mainland China ($n = 1270$), European countries (941), USA (461), Thailand (419) and India (361). At least 27 bacterial genera were represented, with a large fraction of *Klebsiella* and *Escherichia* isolates (2664 and 2154 genomes respectively; Fig. 1b and Supplementary Data 1). Collection dates were recorded for 4816 samples (78.25%). Of these, the majority were collected between 2014 and 2019 (71.05%, Fig. 1c). The dataset also includes 55 genomes collected in 2010 or earlier. These include the 2008 *K. pneumoniae* isolate from Sweden in which $bla_{NDM}$ was first characterized[2]; one 2008 *Enterobacter hormaechei* isolate from India[38]; one 2008 *S. enterica* isolate from London, UK[39]; one *A. baumannii* isolate from an individual of Balkan origin collected in Germany in 2007[40,41]; and nine assembled *E. coli* genomes from urine samples collected in Greece in 2007 (Supplementary Data 1).

The dataset contained 17 known variants of NDM. NDM-1 was the most abundant ($n = 4127$; Supplementary Fig. 3a) with NDM-5 ($n = 2394$) increasing in prevalence after 2012 (Supplementary Fig. 2b, c). Variants showed different associations with plasmid types (Supplementary Fig. 2d) and genera

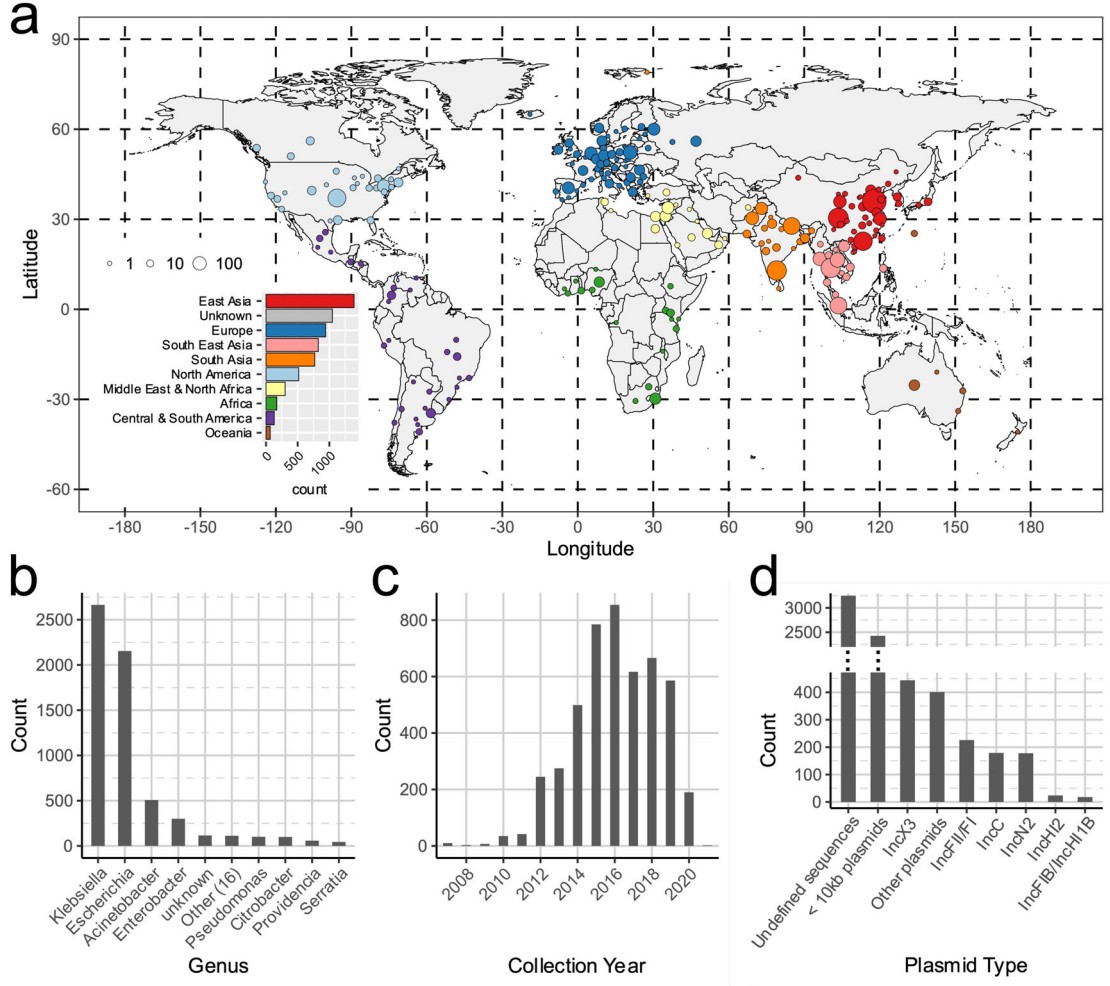

**Fig. 1 Composition of the global dataset of 6155 NDM-positive samples. a** Geographic distribution of $bla_{NDM}$-positive assemblies. Points are coloured by geographic region and their size reflects the number of samples they encompass. The world map was rendered from coordinates provided in *rworldmap*[91] package in R. **b** Distribution of host bacterial genera of NDM-positive samples. **c** Distribution of sample collection years. **d** Distribution of contigs according to the plasmid backbone.

(Supplementary Fig. 2e) but were fairly evenly distributed across the world except for $bla_{NDM-4}$-carrying isolates largely collected in Southeast Asia and $bla_{NDM-9}$ predominantly found in East Asia (Supplementary Fig. 2f).

**Plasmid backbones carrying $bla_{NDM}$.** We identified 33 different replicon types on 1222 contigs using PlasmidFinder[42] (Fig. 1d). The most prevalent replicon type was IncX3 (444 contigs), and abundant types exhibited geographic structure (Supplementary Fig. 4). To further identify uncharacterized plasmid sequences, we mapped 3599 contigs to a set of complete plasmid reference sequences after discarding short contigs (see Methods). This revealed 181 clusters of similar putative plasmid sequences (Fig. 2 and Supplementary Data 2). Most clusters ($n = 105$) grouped contigs of the same replicon type and contained a small number of contigs (only 27 clusters included >10 contigs), in line with a diverse and dynamic population of plasmid backbones for $bla_{NDM}$.

The majority (n = 2427; 68.4%) of $bla_{NDM}$-carrying contigs were associated with small putative plasmids (<10 Kb; Fig. 2). While this could suggest small plasmids play a key role as $bla_{NDM}$ carriers, this pattern could also result from consistently fragmented de novo assemblies due to duplicated ISs and transposons. Consistent with this latter hypothesis, 610 contigs

mapped to pKP-YQ12450 that is likely a 7.8-Kb fragment of a larger plasmid[21]. Conversely, Roach et al. provide evidence that other small $bla_{NDM}$-carrying plasmids (Peruvian pKP-NDM-1_isoforms 1–5) are inherited by descent and are a result of transposon-mediated plasmid fusion[43].

**Resolving structural variants in the $bla_{NDM}$ flanking regions.** To go beyond a static reference-based view of variation around $bla_{NDM}$ and gain a detailed overview of the possible events in its evolution, we developed an alignment-based approach to progressively resolve genomic variation moving upstream or downstream from the gene (see Methods, Fig. 3). In brief, a pairwise discontiguous Mega BLAST search (v2.10.1+)[44,45] is applied to all $bla_{NDM}$-carrying contigs to identify all possible homologous regions between each contig pair. Only BLAST hits covering the complete $bla_{NDM}$ gene are retained (Fig. 3a). Next, starting from $bla_{NDM}$, a gradually increasing 'splitting threshold' is used to monitor structural variants as they appeared upstream or downstream of the gene. At each step, a network of contigs (nodes) that share a BLAST hit with a minimum length as given by the 'splitting threshold' is assessed (Fig. 3b). As we move upstream or downstream and further away from the gene, the network starts to split into smaller clusters, each carrying contigs that share an uninterrupted stretch of homologous DNA, which

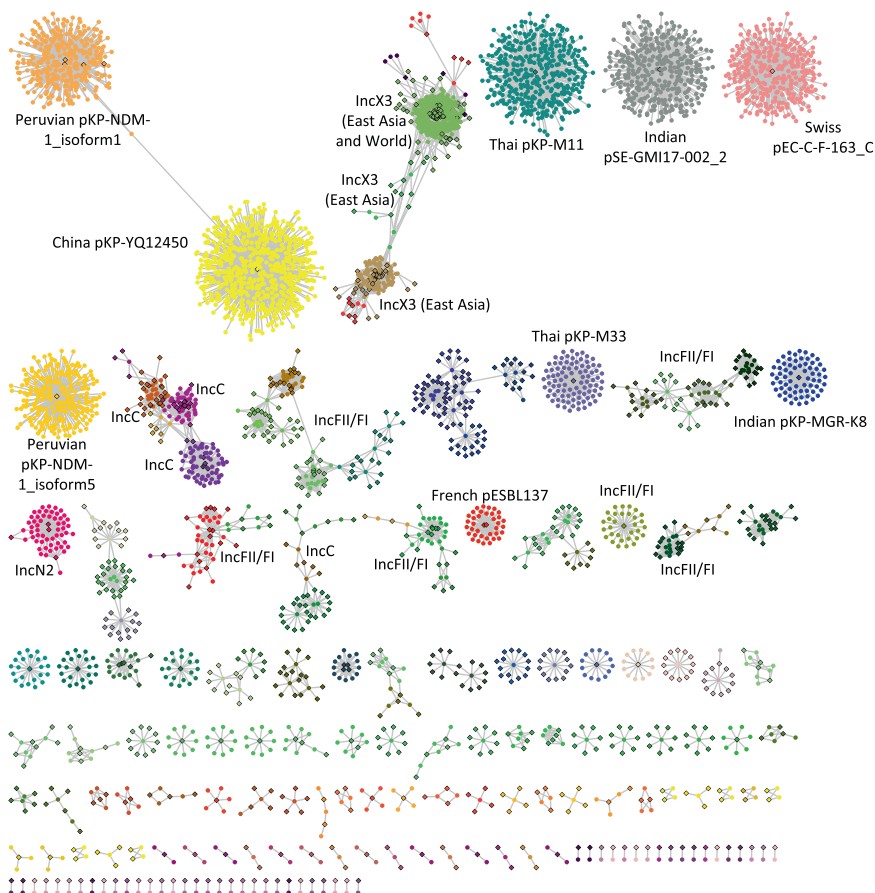

**Fig. 2 A network of $bla_{NDM}$-carrying contigs (circles) mapping to the bacterial plasmid reference sequences (diamonds).** The network is visualized using Cytoscape and coloured according to communities identified by the Infomap algorithm. The largest communities are annotated according to the predominant plasmid type or the reference plasmid. Plasmids of <10 Kb in length include China pKP-YQ12450 ($n = 610$ contigs), Thai pKP-M11 ($n = 399$), Indian pSE-GMI17-002_2 ($n = 354$), Swiss pEC-C-F-163_C ($n = 324$), Peruvian pKP-NDM-1_isoform1-4 ($n = 318$), Peruvian pKP-NDM-1_isoform5 ($n = 226$), Thai pKP-M33 ($n = 91$), Indian pKP-MGR-K8 ($n = 66$), and 39 other <10 Kb putative plasmids.

can be represented as a tree (Fig. 3c). This approach treats the upstream and downstream flanking regions separately rather than simultaneously and is agnostic to whether splitting into 'sequence clusters' is caused by structural variants of the same genomic background or different genomic backgrounds.

Upstream of $bla_{NDM}$, >98% of sufficiently long contigs included a ~75 bp fraction of ISAba125, supporting Tn125 as an ancestral transposon of the $bla_{NDM}$ gene in agreement with previous work[1,5,9,10] (Supplementary Figs. 5 and 6). However, the homology of the region upstream of $bla_{NDM}$ falls quickly: within a few hundred base pairs of the $bla_{NDM}$ start codon the region splits into multiple structural variants, none of which dominate the considered pool of contigs (Supplementary Figs. 5 and 6). We identified 141 different structural variants within 1200 bp upstream of $bla_{NDM}$. This upstream region contained a high number of ISs (e.g., ISAba125 [$n = 243$], IS5 [$n = 426$], IS3000 [$n = 60$], ISKpn14 [$n = 55$], and ISEc33 [$n = 147$]). This transposition hotspot probably contributes to fragmented assemblies: 2269 contigs were excluded from further analysis for being too short (Supplementary Fig. 5).

The downstream flanking region exhibits more gradual structural diversification than the upstream region, with one dominant putative ancestral background (Fig. 4). As illustrated by the stem of the tree of structural variants, many of the 7014 contigs analyzed contained complete sequences of the same set of genes: *ble* (6863 contigs), *trpF* (6038), *dsbD* (5551), *cutA* (2731), *groS* (2175), *groL* (1631). When restricted to $bla_{NDM}$-positive contigs of sufficient

length to possibly harbour the full repertoire of these genes ($n = 3786$), almost half carry all of them ($n = 1631$; 43.1%). In addition, we find dominant structural variants associated with various source databases and sequence lengths hence diminishing the impact of the sampling bias (Supplementary Fig. 7).

**Early events in the spread of $bla_{NDM}$.** While we did not observe any strong overall signal in the distribution of associated plasmid backbones, bacterial genera, or sampling locations, closer examination of mobilome features common to sufficiently large numbers of isolates indicated early events in the spread of $bla_{NDM}$. The putative ancestral Tn125 background, with an uninterrupted downstream ISAba125 element, was seen in contigs mainly from *Acinetobacter* and *Klebsiella* (Fig. 4 top). Conversely, the diversity of bacterial genera carrying ISAba125 upstream is more substantial (Supplementary Fig. 5 top). Only 203 contigs carried a complete ISAba125 downstream of $bla_{NDM}$, of which 147 carried an ISAba125 sequence in proximity (<9 Kb) to the $bla_{NDM}$ start codon. These account for a minority (7%; 147/ 2097) of isolates when sufficiently long contigs are considered. This supports the initial dissemination of $bla_{NDM}$ by Tn125 to other plasmid backbones predominately being mediated by *Acinetobacter* and *Klebsiella*, after which the transposon was disrupted by other rearrangements.

IS3000, both upstream and downstream, was almost exclusively associated with samples from *Klebsiella* (Fig. 4 and Supplementary Fig. 5). Thus, as suggested by Campos et al.[24], Tn3000—a

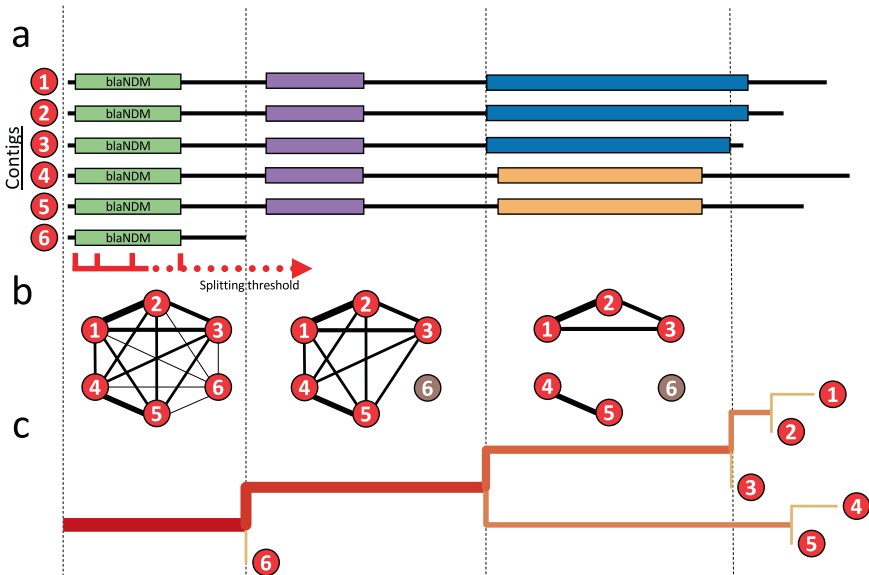

**Fig. 3 Schematic representation of the tracking algorithm splitting structural variants upstream or downstream of bla$_{NDM}$ gene. a** A pairwise BLAST search is performed on all NDM-positive contigs. Starting from *bla*$_{NDM}$ and continuing downstream or upstream, the inspected region is gradually increased using the 'splitting threshold'. **b** At each step, a graph is constructed connecting contigs (nodes) that share a BLAST hit with a minimum length as given by the 'splitting threshold'. Contigs that have the same structural variant at a certain position of the threshold belong to the same graph component, while the short contigs are singled out. **c** The splitting is visualized as a tree where branch lengths are scaled to match the position within the sequence, and the thickness and the colour intensity of the branches correspond to the number of sequences carrying the homology. For a more detailed explanation of the algorithm please refer to the Methods section.

composite transposon made of two copies of IS*3000*—likely remobilized *bla*$_{NDM}$ following the 'fossilization' of Tn*125*; our findings suggest this secondary mobilization primarily happened in *Klebsiella* species. Tn*5403* was found extensively associated with IncN2 plasmids (Fig. 4) that could have placed *bla*$_{NDM}$ in this background via cointegrate intermediate as previously suggested by Poirel et al.[9]. Some elements of the mobilome were geographically linked, e.g., IS*5* that was predominantly found upstream of *bla*$_{NDM}$ on IncX3 plasmids in *Escherichia* from East Asia (Supplementary Fig. 5). IS*5* is known to enhance transcription of nearby promoters in *E. coli*[46] and its abundance and positioning just upstream of *bla*$_{NDM}$ suggests a similar role in this case.

One of the most commonly identified transposable elements in the downstream flanking region (~30% prevalence) was ISCR1 (IS*91* family transposase) (Fig. 4) always accompanied by *sul1* and occasionally in configuration with *ant1* or *pspF*, *ampR*, and *dap* genes. In some cases, a small and possibly fragmented putative IS, which we refer to as 'IS-?', is found further downstream. IS-? bears little similarity to known ISs and it is unclear what role it plays in the mobility of *bla*$_{NDM}$. ISCR1 is found at various positions downstream of *bla*$_{NDM}$ and often in *Escherichia* and *Klebsiella* species. We note that, in most cases, the orientation of ISCR1 should prevent this element from mobilizing *bla*$_{NDM}$ (Fig. 4)[13]. Nevertheless, the prevalence of this element could be due to the several AMR genes it can mobilize, such as *sul1* or *ampR*. ISCR1s are mainly found in complex class 1 integrons[13]; however, not many annotated integrase genes are located within the vicinity of *bla*$_{NDM}$. In fact, only 15 contigs were found to have an integrase <50 Kb away from *bla*$_{NDM}$ and none showed any consistency in integrase placement with respect to *bla*$_{NDM}$. This suggests integrases play a minor role in the dissemination of *bla*$_{NDM}$.

Another notable ISCR element is ISCR27 that is consistently found immediately downstream of the *groL* gene at high prevalence (33.1% of sufficiently long contigs; Fig. 4). Contrary

to its ISCR1 relative, ISCR27 is correctly oriented to mobilize *bla*$_{NDM}$ as is presumed to have happened during the initial mobilization of the progenitor of *bla*$_{NDM}$[10]. However, we find no evidence of subsequent ISCR27 mobility. The origin of RC replication of ISCR27 (*oriIS*; GCGGTTGAACTTCCTATACC) is located 236 bp downstream of the ISCR27 transposase stop codon. The region downstream of this stop codon in all structural variants bearing a complete ISCR27 is highly conserved for at least 750 bp (Fig. 4).

**Subsequent rearrangements dominated by IS26.** Three sharp drops in the number of considered contigs at particular distances downstream of *bla*$_{NDM}$ (see Fig. 4, e.g., region 3000–3300 bp) prompted us to investigate these distinct cut-offs. We mapped 781 raw Illumina paired-end sequencing reads from our dataset back to their matching *bla*$_{NDM}$ contigs. The read overhangs (≥50 bp) that mapped to the downstream end of the contigs were screened against the ISFinder database[47]. The ≥50 bp overhangs associated with 3000–3300 long flanks downstream of *bla*$_{NDM}$ corresponding to the largest observed drop almost exclusively match the left inverted repeat of the IS*26* sequence (Supplementary Fig. 8). Another hotspot associated with IS*26* was found around 7500 bp, while at around 7800 bp, a number of over-hanging reads mapped to IS*Aba125*. These positions roughly match the third drop in the number of contigs observed 7500–8000 bp downstream of *bla*$_{NDM}$. No ISs were found to match the second drop in number of contigs (5000–5250 bp).

IS*26*, although often found in two adjacent copies forming a seemingly composite transposon, is a so-called pseudo-composite (or pseudo-compound) transposon[48]. In contrast to composite transposons, a fraction of DNA flanked by the two IS*26* is mobilized either via cointegrate formation or in the form of a circular translocatable unit (TU), which consists of a single IS*26* element and a mobilized fraction of DNA, and inserts preferentially next to another IS*26*[48,49]. Taken together, the presented results, including Supplementary Fig. 8, suggest three possible explanations for the

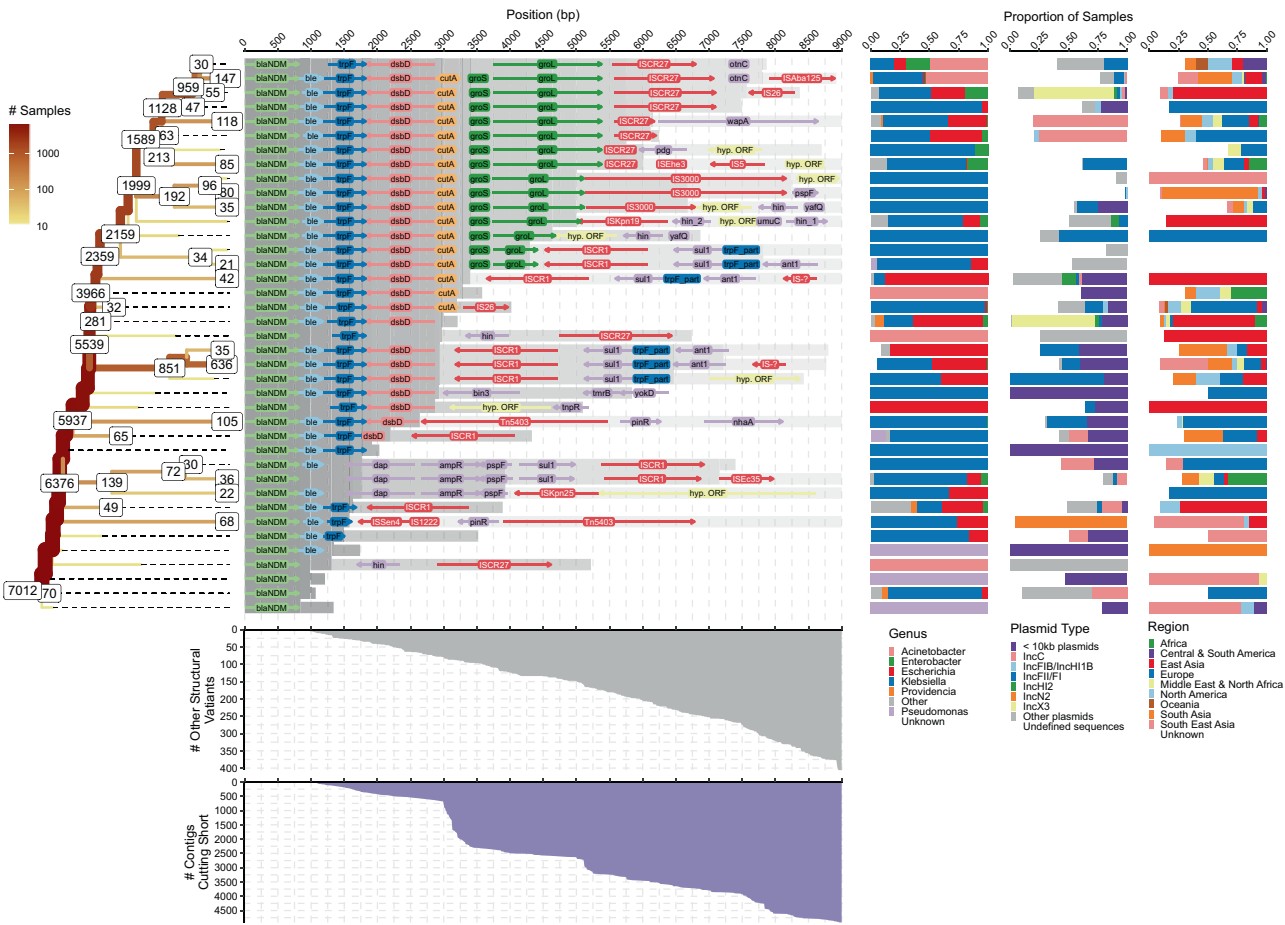

**Fig. 4 Splitting of structural variants downstream of $bla_{NDM}$.** The 'splitting' tree for the most common (i.e., ≥10 contigs) structural variants is shown on the left-hand side. The labels on the nodes indicate the number of contigs remaining on each branch. Labels of (yellow) branches with <20 contigs are not shown. The other contigs either belong to other structural variants or were removed due to being too short in length. The number of contigs cutting short is indicated by the area chart at the bottom. Similarly, the number of less common structural variants is indicated by the upper area chart. Genome annotations provided by the Prokka and Roary pipelines of the most common structural variants are shown in the middle of the figure. The homologous regions among structural variants are indicated by the grey shading. Some of the structural variants and branches were intentionally cut short even though their contigs were of sufficient size or longer. This was done to prevent excessive bifurcation and to make the tree easier to interpret. In particular, branches with more than 75% of contigs lost due to variation and short length were truncated. The distribution of genera, plasmid backbones and geographical regions of samples that belong to each of the common structural variants is shown on the right-hand side.

presence of short $bla_{NDM}$ carrying contigs in the dataset: (i) the presence of IS26 TUs in the host cell; (ii) other circular DNA formations mediated by plasmid recombination, transposons[9,43] or ISCR elements[12,50]; (iii) missassembly of contigs due to the presence of multiple copies of the same ISs[51].

To further investigate the mobility of $bla_{NDM}$, we characterized the most common (pseudo-)composite transposons theoretically capable of mobilizing $bla_{NDM}$ (Fig. 5). These were defined as stretches of DNA flanked by two matching complete or partial ISs <30 Kb apart and enclosing $bla_{NDM}$. In total, we identified 640 composite transposons in 468 contigs that comprised 31 different types with the most frequent being: IS26 (231 instances), IS3000 (forming Tn3000; 168), ISAba125 (forming Tn125; 138 instances), and IS15 (28) (Fig. 5b). Interestingly, we observe 80 cases where >2 of the same IS flank $bla_{NDM}$. These are mostly IS26 (59) that could indicate the presence of cointegrate formation[48] and showcase increased activity of this particular insertion element. Only 431 of the 640 putative composite transposons identified contained both complete flanking ISs, while others had at least one IS partially truncated. In addition, 1681 ISCR27, and 150 ISCR1 were found in similar proximity and appropriate orientation to mobilize $bla_{NDM}$ (Fig. 5b).

However, as mentioned earlier, their role in the transposition of $bla_{NDM}$ appears minor.

In the majority of cases, composite transposons Tn125 and Tn3000 were found to have a consistent length ranging from 7 to 10 Kb (Fig. 5a). Similarly, ISCR1 and ISCR27 are found at fixed positions downstream of $bla_{NDM}$. However, the lengths of transposons formed by IS15, a known variant of IS26[52], and especially IS26 were found to be more variable. Pairs of IS26 are found to be 2.5-30 Kb apart again consistent with increased activity and multiple independent insertions. We note that IS15 and IS26 occur at increased presence in samples collected in East and Southeast Asia (Fig. 5c). These occur roughly equally in *Escherichia* and *Klebsiella* genera (Fig. 5d) and are associated with multiple plasmid backbones, but predominantly on IncF plasmids (Fig. 5e). Tn125 and Tn3000 have a notable predominance in the Indian subcontinent (Fig. 5c) and largely in the *Acinetobacter* and *Klebsiella* genera, respectively (Fig. 5d).

**Molecular dating of key events.** We estimated the relative timing of the formation of the Tn125 and Tn3000 transposons (see Methods). After selecting only contigs with conserved transposon configurations we aligned each transposon region and identified

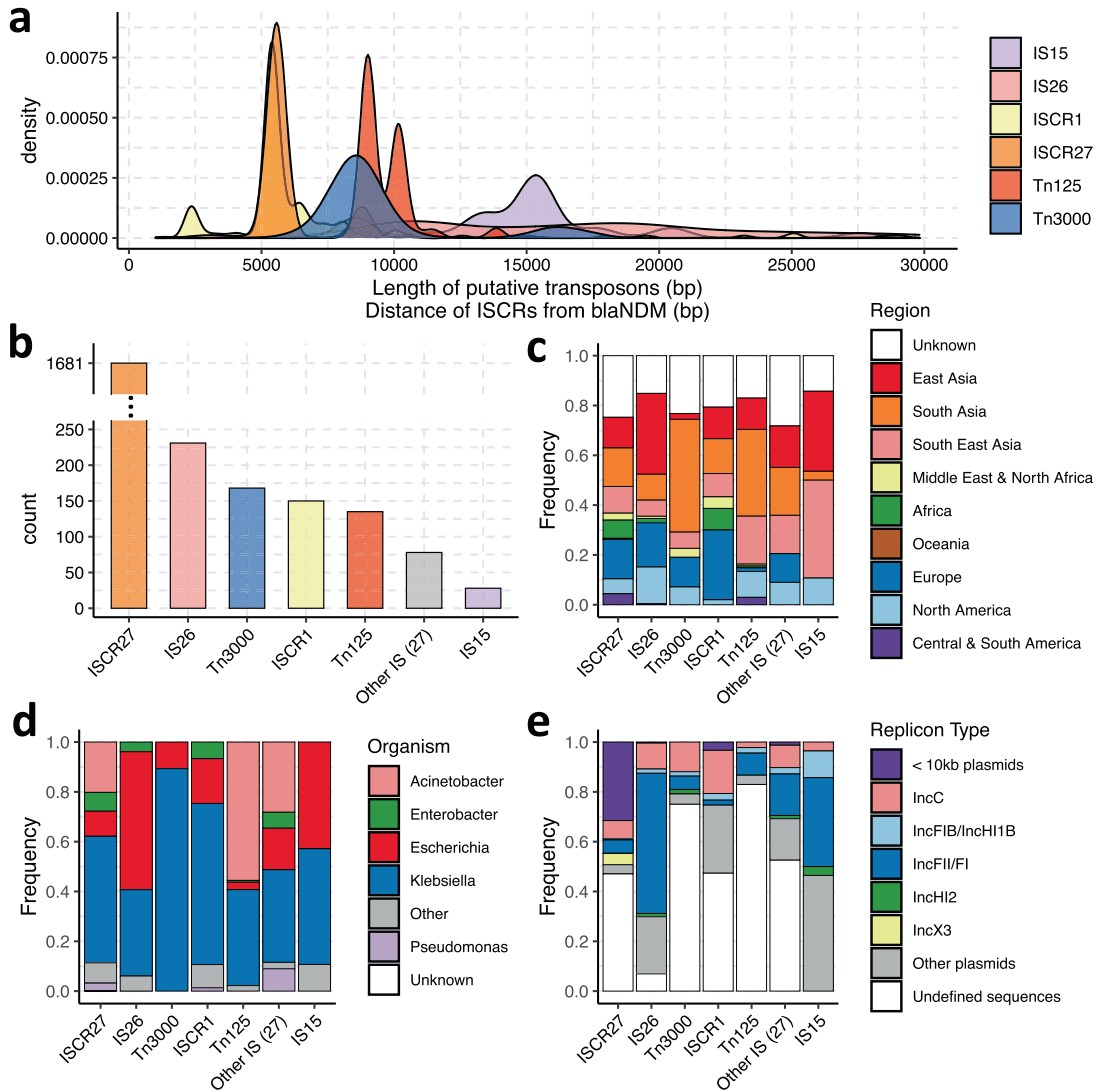

**Fig. 5 Global prevalence and genetic context of most frequent putative (pseudo-) composite transposons and insertion sequences containing common regions (ISCRs) capable of mobilizing *bla*<sub>NDM</sub> gene.** Transposons were defined as stretches of DNA flanked by two matching complete or partial ISs <30 Kb apart and enclosing *bla*<sub>NDM</sub>. Putative pseudo-composite transposons were labelled according to their constituent ISs (IS15 and IS26). **a** Marginal distributions of transposon lengths or distances of ISCRs from *bla*<sub>NDM</sub> start codon. **b** Overall counts of the frequent transposable elements (i.e., >25 representatives). **c–e** Bar plots, respectively, indicating proportions of plasmid backbones, bacterial genera and sampling location associated with most frequent transposable elements.

the likely root (i.e., ancestral) sequence by assessing temporal patterns (Supplementary Figs. 9 and 10; see Methods). Overall, we observed fewer SNPs, mostly located within the transposase gene, in the alignment of Tn*3000* compared to Tn*125*, but observed a notable temporal signal for both (Supplementary Figs. 11 and 12). We also assessed temporal signal for three other prevalent insertion events (Fig. 4), namely: *bla*<sub>NDM</sub> with downstream ISCR27, *bla*<sub>NDM</sub> with correctly oriented downstream *folP*-ISCR1 (+strand), and *bla*<sub>NDM</sub> – dsbD with downstream ISCR1 (− strand) ending with an unknown putative IS (labelled IS-?). However, no significant temporal signal was recovered for these events.

The Bayesian analysis indicated that the most recent common ancestor (MRCA) of the Tn*125* transposon carrying the *bla*<sub>NDM</sub> gene dated to before 1990 (Fig. 6a). While the time intervals are uncertain, the results are consistent with an MRCA in the mid-twentieth century—strikingly half a century prior to the first reported Tn*125*-*bla*<sub>NDM</sub>-positive isolates[7]. Conversely,

the mobilization of *bla*<sub>NDM</sub> by Tn*3000* is estimated to have happened later at the turn of the millennium (Fig. 6b). These findings are consistent with a wider narrative whereby the spread of *bla*<sub>NDM</sub> was initially driven by Tn*125* mobilization before subsequent transposition by Tn*3000*, IS26 and others.

**Temporal diversity in *bla*<sub>NDM</sub> isolates suggests the role of plasmids.** The earliest samples in our dataset are from 2007 to 2010 and comprise 21 *bla*<sub>NDM</sub>-positive isolates. These already encompass seven bacterial species, collected in eight countries spanning four geographic regions (17 clinical samples and four of unknown origin from South Asia, Middle East, Oceania, and Europe). Such a wide host and geographic distribution, even in the earliest available genomes, illustrates the extraordinarily high mobility of *bla*<sub>NDM</sub> at this stage and is consistent with our molecular dating estimates.

In order to trace the progress of *bla*<sub>NDM</sub>'s rapid spread after 2005 (coinciding with the first published observations), we measured

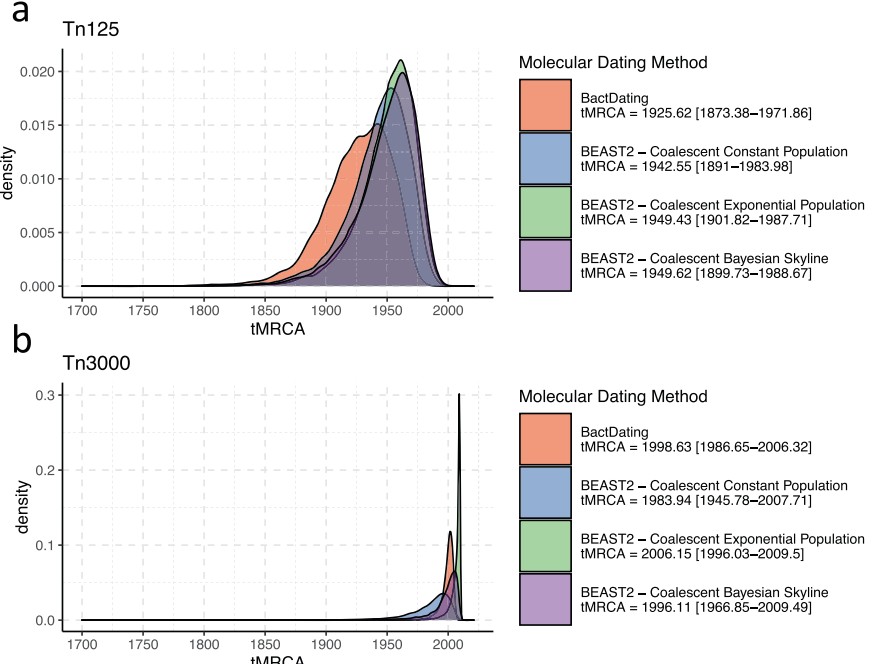

**Fig. 6 Molecular dating of *bla*_NDM mobilization by Tn125 and *Tn3000*.** Posterior density distributions of ancestral sequence age (i.e., root height) for the *Tn125* (**a**) and *Tn3000* (**b**) transposons. The ancestral sequence emergence was estimated using two Bayesian tip-dating approaches implemented in BactDating and BEAST2. Three different population growth priors were used in case of BEAST2: Coalescent Constant Population, Coalescent Exponential Population, and Coalescent Bayesian Skyline as given by the colour scheme and legend at right. Median estimates with 95% highest density interval (HDI) are provided in the panel legends.

diversity over time for several metadata categories, including country, genera, plasmid backbone and IS presence (Supplementary Fig. 15; see Methods). The change in the diversity of the countries associated with *bla*_NDM-positive isolates was used to approximate the broad patterns of global dissemination of *bla*_NDM. Our results are consistent with the spread stabilizing between 2013 and 2015, with a gradual decline in diversity afterwards (Supplementary Fig. 15a). This observation supports a scenario whereby the global dissemination of NDM took place over 8–10 years. Temporal diversity of bacterial genera was largely unchanged, consistent with *bla*_NDM having been highly mobile across genera since at least 2010 (Supplementary Fig. 15b).

The estimated change in the diversity of countries associated with *bla*_NDM-positive isolates was positively correlated with other metadata categories (Supplementary Fig. 16) suggesting it holds information that can be leveraged to reconstruct dissemination trends. The strongest correlation was found between the diversity of countries and plasmid backbones ($\rho = 0.864$ [0.691–0.964]) supporting a strong dependence between the two (Supplementary Fig. 16b). To further investigate this relationship, we assessed the correlation between genetic and geographic distance between pairs of confirmed plasmid contigs (tested for IncF, IncX3, IncC, IncN2 and confirmed plasmid contigs >10 Kb) as a function of the distance downstream of *bla*_NDM gene (Supplementary Fig. 17, see Methods).

No relationship was detected for IncX3 and IncN2 plasmids (Supplementary Fig. 17a, b) likely due to the lack of long plasmid sequences and deficient geographic distance between pairs of plasmids as both replicon types are mostly localized to China and India respectively (Supplementary Fig. 4). However, in all other cases aside from IncN2 plasmids, a peak in the correlation recovered between genetic and geographical distance was observed immediately downstream of *bla*_NDM possibly signifying more recent and local genome reshuffling events (Supplementary Fig. 16). More importantly, in IncF and IncC, and other

confirmed plasmid contigs, a notable and gradual increase in the strength of correlation was noted further downstream as more plasmid sequence is included in the analysis (Supplementary Fig. 17b–d). These trends suggest that plasmids carrying *bla*_NDM are geographically structured and that dissemination of *bla*_NDM is a fundamentally spatial process. This would be consistent with the existence of plasmid niches: settings to which particular plasmids are more adapted.

## Discussion

In this study, we have characterized the extant structural variation around *bla*_NDM in a large global dataset in order to reconstruct its evolutionary history and the main actors underlying its spread. Our results highlight an ancestral background of *bla*_NDM as well as several insertion events and a myriad of other genetic reshuffling, together pointing to an early emergence of *bla*_NDM followed by more recent and rapid dissemination globally. Genetic reshuffling and mobilization of *bla*_NDM by multiple transposons aided its rapid dissemination via a multitude of plasmid backbones.

We go beyond previous smaller studies by dating the MRCA of the hypothesized ancestral form—the transposon Tn125, together with *bla*_NDM in its chimeric form[10]—to pre-1990, and possibly well back into the mid-twentieth century. A likely scenario is an origin in *Acinetobacter* in the Indian subcontinent. We note that Tn125 is mostly present in *Acinetobacter* and *Klebsiella* species and it is likely this transposon played an important role in early plasmid jumps of *bla*_NDM, given it is the dominant transposon in our comprehensive dataset that encompasses the ancestral genetic background of *bla*_NDM—*groS/groL* genes and ISCR27 sequence. We also estimate the formation of a secondary transposon, involving *Tn3000*, which remobilized the region likely in *Klebsiella* species sometime between the 1980s and early 2000s. However, we suggest *Tn3000* likely played a lesser role in the

early spread of $bla_{NDM}$ as it does not include the ISCR27 found at least partially preserved in many samples.

In total, 31 different putative transposons were identified within our dataset. Their role, together with integrons and other transposable elements, is likely mostly minor or disruptive, as suggested for ISCR1. However, we do identify IS26 as of interest, given it frequently forms putative transposons in our dataset, especially in IncF plasmids. IS26 is known for its increased activity and rearrangement of plasmids in clinical isolates[53] and has been observed to drive within-plasmid heterogeneity even in a single *E. coli* isolate[54]. Thus, IS26-flanked pseudo-composite transposons likely represent the most important contributor to the genetic reshuffling of $bla_{NDM}$ in recent times.

Our assessment of temporal diversity of countries of origin of $bla_{NDM}$ positive isolates supports a globalization peak in 2013–2015. Such a rapid 8–10-year worldwide spread has been suggested for other important mobile resistance genes such as the *mcr-1* gene, mediating colistin resistance[55]. The extent to which this model of 'rapid global spread' applies to other transposon-borne resistance elements remains to be determined.

We found 33 different plasmid types carrying $bla_{NDM}$ and a positive correlation between genetic distance calculated for differing lengths of plasmid backbones and geographic distances of sampling locations. This observation is consistent with the existence of a constraint on plasmid spread, i.e., plasmid niches, that may exist as a result of local ecological and evolutionary pressures acting on particular plasmid backbones. Such forces may include country boundaries limiting population movement, region-specific patterns in antibiotic usage, influence of co-resistance, plasmid fitness costs, conjugation rates and copy numbers, the narrow host range of the majority of bacterial plasmids[56], or plasmids being associated with particular locations or environmental niches[57], all may contribute to restricting plasmid geographical range. Thus, an introduction of another plasmid into a foreign plasmid niche may lead to plasmid loss or fast adaptation by, for instance, acquisition of resistance and other accessory elements. This hypothetical scenario also provides an opportunity for resistance to spread by transposition or recombination, by which a new resistance gene could establish itself into another plasmid niche. In the case of $bla_{NDM}$, this would also imply that after the initial introduction of $bla_{NDM}$ to a geographic region, dissemination and persistence of the gene could proceed idiosyncratically—selection for carbapenem resistance being just one of many selective pressures acting on plasmid diversity.

The methodological framework we developed reconstructs the immediate up- and downstream backgrounds of the blaNDM gene separately, which is a limitation inherent to our network-based algorithm. In the case of $bla_{NDM}$, we deem this approach to be satisfactory since the downstream region is dominated by a single putative ancestral background while the alignment rapidly breaks down upstream. For future applications to other mobile genetic elements, it may be worthwhile to jointly resolve genomic variation in both directions. The solution to this problem using the presented methodology does not appear straightforward or computationally efficient. A simple satisfactory alternative would be to select an anchorpoint falling upstream or downstream of the genetic element of interest, rather than starting from the gene of interest as we did in the present study.

The importance of transposon movement has been previously demonstrated by work on plasmid networks[56,58], as well as several papers promoting a Russian-doll model of resistance mobility[55,59]. Considering our results, we suggest a conceptual framework of AMR gene dissemination across genera where plasmid mobility is for the most part restricted. Although plasmids can facilitate rapid spread within species and geographical regions, plasmid transfer is not the main driver of widespread dissemination. Instead, most plasmid horizontal transfers are likely only transient, with plasmids generally failing to establish themselves in the new bacterial host, though such aborted plasmid exchanges still provide a crucial opportunity for between-plasmid transposon jumps and genetic recombination to spread AMR genes across bacterial species.

## Methods

**Compiling the curated dataset of $bla_{NDM}$ sequences.** We compiled an extensive dataset of 6155 bacterial genomes carrying the $bla_{NDM}$ gene from several publicly available databases. A total of 2632, 1158 and 1379 fully assembled genomes were downloaded from NCBI Reference Sequence Database[33,60] (RefSeq; accessed on 15 April 2021), NCBI's GenBank[34] (accessed on 15 April 2021), and EnteroBase (accessed on 27 April 2021)[35], respectively. The EnteroBase repository was screened for $bla_{NDM}$ using BlastFrost (v1.0.0)[61] allowing for one mismatch. In addition, we used the Bitsliced Genomic Signature Index (BIGSI) tool (v0.3)[62] to identify all SRA unassembled reads that carry the $bla_{NDM}$ gene. At the time of writing, a publicly available BIGSI demo did not include sequencing datasets from after December 2016. Therefore, we manually indexed and screened an additional 355,375 SRA bacterial sequencing datasets starting from January 2017 to January 2019. We required the presence of 95% of $bla_{NDM-1}$ k-mers to identify NDM-positive samples from raw SRA reads. This led to the inclusion of a further 882 isolates. The dataset also included 104 NDM-positive genomes from 79 hospitalized patients across China and 25 livestock farms selected from two previous studies[63,64]. These were sequenced using paired-end Illumina (Illumina HiSeq 2500) and PacBio (PacBio RS II). The sequencing reads are available on the Sequence Read Archive (SRA) under accession number PRJNA761884. All reads were de novo assembled using Unicycler (v0.4.8)[65] with default parameters while also specifying hybrid mode for those isolates for which we had both Illumina short-read and PacBio long-read sequencing data. Spades (v3.11.1)[66] was applied, without additional polishing, for cases where Unicycler assemblies failed to resolve.

Assembled genomes were retained when they were derived from a single BioSample identifier. Contigs carrying the $bla_{NDM}$ gene were identified using BLAST (v2.10.1+)[44]. Forty-eight contigs were found to carry more than one copy of $bla_{NDM}$ and were not included in our analyses and 88 samples were excluded due to having partial (<90%) $bla_{NDM}$ hits. Fourteen assemblies had a single $bla_{NDM}$ gene split into two contigs; these 28 contigs were also excluded. Several contigs were also removed due to poor assembly quality. The filtering resulted in a dataset of 7148 contigs (6155 samples) that were used in all subsequent analyses. Of these, 958 assembled genomes were found to contain $bla_{NDM}$ on multiple (mostly two) contigs, each harbouring a single and complete copy of $bla_{NDM}$. Even though the information about sequencing platform or assembly methods of most samples from RefSeq, GenBank and Enterobase databases could not be determined, the distribution of $bla_{NDM}$-positive contig lengths (Supplementary Fig. 1) reveals they are likely to be based on short reads with the minority of contigs, mostly from RefSeq, reaching the quality of a hybrid de novo assembly. The full table of contigs and metadata considered is available as Supplementary Data 1.

**Quality assessment of SRA $bla_{NDM}$-positive contigs.** Quality assessment was performed on all $bla_{NDM}$-positive contigs assembled from samples obtained from SRA database. The phred quality scores and mapping coverage was obtained with BBMap tools[67]. Overall, the $bla_{NDM}$-positive contigs showed good mapping coverage predominantly exceeding per base coverage of 30× (Supplementary Fig. 3a). Similar coverage was obtained for SRA contigs used in molecular dating (Supplementary Fig. 3b). Phred score was used to assess the quality of SNP calls found in SRA contigs used in molecular dating analysis. A total of three variable positions (i.e., SNPs) were found within 30 SRA contigs compared to the inferred ancestral sequence (Supplementary Fig. 2c, d). In ten contigs, two out of these three variable positions were found, to have slightly lower however still acceptable (>20), Phred quality score.

**Annotating the dataset.** Full metadata for each genome was collected from its respective database and the R package 'taxize'[68] was used to retrieve taxonomic information for each sample. In the case of samples for which exact sampling coordinates were not provided, Geocoding API from Google cloud computing services was used to retrieve coordinates based on location names.

Coding sequences of all NDM-positive contigs were annotated using the annotation tool Prokka (v1.14.6)[69] and Roary (v3.13.0)[70] run with minimum blastp percentage identity of 90% (-i 0.9) and disabled paralog splitting (-s). To identify plasmid replicon types[71], contigs were screened against the PlasmidFinder replicon database (version 2020-02-25)[42] using BLAST (v2.10.1+)[44] where only BLAST hits with a minimum coverage of 80% and percentage identity of ≥95% were retained. In cases where two or more replicon hits were found at overlapping positions on a contig, the one with the higher percentage identity was retained. Identified plasmid types were used to cluster contigs into broader plasmid groups: IncX3, IncF, IncC, IncN2, IncHI1B, IncHI2 and other (Fig. 1d).

NDM-positive contigs were also screened against a dataset of complete bacterial plasmids. Bacterial plasmid references were obtained from RefSeq[33] and curated to

include plasmids from a bacterial host and with a sequence description, which implies a complete plasmid sequence (regular expression term used: 'plasmid.*complete sequence')[72]. Mash, a MinHash based genome distance estimator[73], was applied with default settings to evaluate pairwise genetic distances between contig sequences and plasmid references. Contig-reference hits with less than 0.05 Mash distance and less than 20% difference in length were retained. Additional pruning was implemented such that, for each contig analyzed, only the 10% of best scoring plasmid reference hits were retained. A table of pairwise genetic distances between contigs and references was represented as a network that was then analyzed with the infomap[74] community detection algorithm implemented in *igraph*[75] R package. Contigs were annotated according to their community membership and the network was visualized using Cytoscape[76] (Fig. 2).

**Resolving structural variants of NDM-positive contigs**. A novel alignment-based approach was used to identify stretches of homology (i.e., maximal alignable regions) as well as structural variations across all contigs upstream and downstream of $bla_{NDM}$ gene. A conceptual illustration of the method is presented in Fig. 3. First, contigs carrying $bla_{NDM}$ were reoriented such that the $bla_{NDM}$ gene was located on the positive-sense DNA strand (i.e., facing 5′ to 3′ direction). A discontiguous Mega BLAST (v2.10.1+)[45] search with default settings was then applied against all pairs of retained contigs. This method was selected over the regular Mega BLAST implementation as it is comparably fast, but more permissive towards dissimilar sequences with frequent gaps and mismatches. BLAST hits including a complete $bla_{NDM}$ gene represent maximal stretches of homology around the gene for every pair of contigs. The analysis continues by considering only portions of BLAST hits: (i) the start of $bla_{NDM}$ gene and the downstream sequence or (ii) the end of the $bla_{NDM}$ gene and the upstream sequence depending on the analysis at hand: the downstream or the upstream analysis, respectively. This trimming of BLAST hits establishes $bla_{NDM}$ as an anchor and enables comparisons to be made across all samples.

A table of BLAST hits can be considered as a network (graph), where each pair of contigs (i.e., nodes) are connected by the edge weighted by the length of the BLAST hit. The algorithm proceeds with a stepwise network analysis of BLAST hits. For this purpose, a 'splitting threshold' was introduced. Starting from zero that represents the start/end position of $bla_{NDM}$ gene, the threshold is gradually increased by 10 bp. At each step, BLAST hits with a length lower than the value given by the 'splitting threshold' are excluded. Thus, as the 'splitting threshold' increases, a network of BLAST hits is also pruned and broken down into components—groups of interconnected nodes (contigs). It is expected that contigs within each component share a homologous region downstream (or upstream) of $bla_{NDM}$ at least of the length given by the threshold. It is therefore not possible for a single contig to be assigned to multiple components. Components of size <10 are labelled as 'Other Structural Variants'. Also, contigs that are shorter than the defined 'splitting threshold' and share no edge with any other contig are considered as 'cutting short'. Similar clustering algorithms have been previously described, e.g., CAST[77], but were used for a different application.

By tracking the splitting of the network as the 'splitting threshold' is increased, one can determine clusters of homologous contigs at any given position downstream or upstream from the anchor gene (here $bla_{NDM}$), as well as the homology breakpoint. The precision of the algorithm is directly influenced by the step size, in this case 10 bp, and the alignment algorithm, in this case discontiguous Mega BLAST. We assessed the precision of the algorithm on the tree of structural variations downstream of $bla_{NDM}$ (Fig. 4). To this end, we compared extended 50 bp sequence fragments of each branching point in the tree checking for missed homologies and comparing Mash distances between pairs of branched-out contigs. We found no similarities among 50 bp fragments of any split branches. The described algorithm is available at https://github.com/macman123/track_structural_variants.

**Analyzing the contig overhanging reads**. To investigate the reasons behind a number of distinctively short $bla_{NDM}$-carrying contigs, we mapped 781 raw Illumina paired-end sequencing reads (originally downloaded from SRA) back to their matching contigs. The mapping was done using BBMap[67] (v38.59; *maxindel* = 0 and *minratio* = 0.2 settings). Within the output SAM file, only the overhanging reads with the CIGAR string matching the *[0-9]\*M[0-9]\*S* regular expression were selected. All overhangs of reads ≥50 bp were screened against ISFinder database[47].

**Molecular tip-dating analysis**. The 112 complete Tn*125* and 73 complete Tn*3000* contigs with a known collection date and harbouring $bla_{NDM}$ were sequentially aligned (--pileup flag) using Clustal Omega (v1.2.3)[78] specifying the $bla_{NDM-1}$ sequence (FN396876.1) as a profile. Each alignment was manually inspected using UGENE (v38.0)[79]. Quality assessment (Supplementary Fig. 2) and manual inspection of the alignments did not reveal any error-prone site; thus, no site masking was implemented in molecular tip-dating analysis. The ancestral (i.e., root) sequence was determined by evaluating SNP frequencies over time (Supplementary Figs. 9 and 10). Due to a short sampling time span and relatively few mutations present, it is unlikely that any one non-ancestral SNP has become dominant in the population. Therefore, we expect the ancestral sequence to have a higher SNP frequency in earlier years.

We find that, in all but two cases, the consensus sequence of an alignment displays this behaviour. The first exception is the consensus sequence allele of Tn*125* at the variable position 441 (Supplementary Fig. 9). This allele has a low frequency in 2009. However, by inspecting the allele frequency table, we observe the low frequency is based on a single sample. Leaving out this early sample restores the desired frequency pattern; hence, the consensus allele is considered ancestral in this case. The second exception is the variable position 449 in the case of the Tn*3000* alignment (Supplementary Fig. 10). The consensus allele 'a' is not found in the early sample from 2009. Both allele 't', present in the early sample, and allele 'a' were found equally frequent in more recent samples. Thus, due to lack of other evidence, allele 't' was considered ancestral. Determined ancestral sequences were used to evaluate temporal signal in the alignment, and in the subsequent rooting of phylogenetic trees.

Date randomization (1000 iterations) and linear regression analyses were employed to estimate the presence of temporal signal in the alignment[80–82] (Supplementary Figs. 11 and 12). The analyses did not reveal any obvious outliers or problems with the data in case of both Tn*125* and Tn*3000*.

Bayesian-based molecular dating approaches were implemented in BEAST2 (v2.6.0)[83] and BactDating[84] to infer the date of the emergence of the two transposons. The BactDating analysis was run with *strictgamma* model specification and coalescent prior on the tree. All BEAST2 analyses specified a strict prior on the molecular clock with a uniform distribution from 0 to infinity and the generalized time reversible (GTR) substitution model prior. Three population size models with default settings were used in BEAST2 analyses: Coalescent Constant population, Coalescent Exponential population, and Coalescent Bayesian Skyline. Multiple general coalescent population size models and loose (default) priors were used to corroborate dating results because, to the best of our knowledge, the complexity of transposon mobility is not currently described in any models available in BEAST2. Such an approach has yielded wider, but credible confidence intervals on the dating results.

Strict molecular clock prior was chosen over relaxed clock due to computational cost and relatively recent emergence of $bla_{NDM}$ element, but more importantly due to lack of host structure across transposon phylogenetic trees. In addition, all BEAST2 and BactDating runs were supplied with a maximum likelihood phylogenetic tree (starting tree prior) constructed from both transposon alignments using RAxML (v8.2.12)[85] with specified GTRCAT substitution model and rooted using the inferred ancestral sequences. The chosen MCMC chain lengths for BactDating and BEAST2 runs were $10^7$ and $1.5 \times 10^9$, respectively, to ensure convergence. We evaluated effective sample sizes (ESS) of the posterior distributions using *effectiveSize* function implemented in *coda*[86] R package after discarding the first 20% of burn-in (Supplementary Figs. 13 and 14). All BEAST2 and BactDating runs successfully converged with ESS of the posteriors close to or above 200. BEAST2 input files are available as xml files in Supplementary Data 3.

**Estimating Shannon entropy among NDM-positive contigs**. We estimated Shannon entropy ('diversity') for several categorizations of $bla_{NDM}$-containing contigs: country of sampling, bacterial host genera, plasmid backbones (determined by mapping to plasmid reference sequences), and ISs flanking the $bla_{NDM}$ gene. To estimate entropy of the population and to provide confidence intervals around our estimates, we use bootstrapping with replacement (1000 iterations). At each iteration, entropy was estimated for a sampled set of contigs (X) classified into $n$ unique categories according to the following formula:

$$H(X) = -\sum_{i=1}^{n} P(x_i) \log P(x_i), \tag{1}$$

where the probability $P(x_i)$ of any sample belonging to any particular category $x_i$ (e.g., country or plasmid backbone) is approximated using the category's frequency. Accordingly, higher entropy values indicate an abundance of equally likely categories, while lower entropy indicates a limited number of categories.

**Estimating correlation between genetic and geographic distance**. Geographic distance between pairs of samples was determined using their sampling coordinates and the *geodist*[87] R package. Exact Jaccard distance (JD) was used as a measure of the genetic distance calculated using the tool Bindash (v0.2.1)[88] with $k$-mer size equal to 21 bp. The JD is defined as the fraction of total $k$-mers not shared between two contigs. JD between all pairs of contigs was first calculated on a 1000 bp stretch of DNA downstream of $bla_{NDM}$ start codon continuing with a 500 bp increment. At each increment, the two distance matrices (genetic and geographic) were assessed using the *mantel* function (Spearman correlation and 99 permutations) from the *vegan*[89] package in R. The correlation between genetic and geographic distance, was plotted as a function of distance from $bla_{NDM}$ gene (Supplementary Fig. 15).

**Reporting summary**. Further information on research design is available in the Nature Research Reporting Summary linked to this article.

## Data availability

The accession numbers of bacterial genomes obtained from the RefSeq, Enterobase and SRA databases are given in the Supplementary Data 1. One hundred and four paired-end

Illumina and PacBio sequencing data from China generated in this study are available on SRA database under the BioProject accession number PRJNA761884. Whole-genome *de novo* assemblies are available on GenBank under the same BioProject accession number. Filtered dataset of 7155 $bla_{NDM}$ bearing contigs is available on Figshare (https://doi.org/10.5522/04/16594784). Source Data are provided with this paper.

## Code availability

All software used in this research are listed in Methods. An implementation of the algorithm used to track structural variations around $bla_{NDM}$ is available at GitHub (https://github.com/macman123/track_structural_variants)[90].

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

## Acknowledgements

M.A. was supported for a PhD scholarship from University College London. H.W. is supported by National Natural Science Foundation of China (32141001 and 81625014). L.v.D., H.W. and F.B. acknowledge financial support from the Newton Fund UK-China NSFC initiative (MRC Grant MR/P007597/1 and 81661138006). L.v.D. and F.B. are supported by a Wellcome Institutional Strategic Support Fund (ISSF3) – AI in Health-care (19RX03). F.B. additionally acknowledges support from the BBSRC GCRF scheme and the National Institute for Health Research University College London Hospitals Biomedical Research Centre. L.v.D is supported by a UCL Excellence Fellowship. M.A., L.v.D and F.B. acknowledge UCL Biosciences Big Data equipment grant from BBSRC (BB/R01356X/1). L.P.S. is a Sir Henry Wellcome Postdoctoral Fellow funded by Well-come (Grant 220422/Z/20/Z). The funders had no role in study design, data collection, interpretation of results, or the decision to submit the work for publication. Lastly, M.A. would like to thank Nicola de Maio for informal discussions which led to the idea for the algorithm used to track structural variants.

## Author contributions

M.A., F.B., L.v.D. and H.W. conceived the project and designed the experiments. M.A., L.v.D., L.P.S., and N.L. collected data from online repositories. R.W., Y.Y., Q.W., S.S, and H.C. sequenced samples from Chinese hospitals. M.A., L.v.D. and R.W. de novo assembled all the genomes. M.A. performed all the analyses under the guidance of L.v.D. and F.B. M.A., L.v.D. and F.B. take responsibility for the accuracy and availability of the results. M.A. wrote the paper with contributions from L.P.S., L.v.D. and F.B. All authors read and commented on successive drafts and all approved the content of the final version.

## Competing interests

The authors declare no competing interests.
