## [Peer Review File · Nature Communications]

Reviewers' Comments:

Reviewer #1:

Remarks to the Author:

This revision does not seem to have taken on board any of the comments made about the previous submission and the responses to many of those comments do not actually address them. This is a surprising approach. A few references covering what is already known were included. However, it's not clear what the analysis presented adds to what is known already from those references.

Hence, it still does not increase what is known about the dissemination of the blaNDM gene. The dating story (an addition) fails to account for sequence quality, so how many of the SNP are simply errors and how many real is not addressed. This invalidates this aspect of the work.

In addition, the methodology is unnecessarily complicated. For example, the contig containing Tn125 can be found simply in draft genomes using BLAST searches. Overall, the findings lag way behind what is already known. Splitting information about the context upstream from the context downstream is a serious concern as the complete picture requires every pair to be linked and other available Methods supply this information.

Overall the text was essentially hearsay as how the findings were reached is not adequately covered. For example, the plasmid type can be identified if the plasmid sequence is complete but generally it will not be possible to deduce even plasmid association from draft genomes. How these issues were dealt with is not mentioned or explained. How many had a contig containing both the NDM gene and identifiable plasmid sequence?

It is well known that some sequencing technologies are more error prone than others. To look in detail at SNPs, the poor-quality genomes must first be filtered out. Even then, there are problems as conjugative transfer can fix and then amplify a sequence variation.

The text in multiple Figures is too small or of insufficient contrast with the background to be read easily.

Reviewer #2:

Remarks to the Author:

As I have stated in my first review, this is not a novel methods paper, therefore I will not further pursue methodological novelty in my assessment. Just a couple of comments to clarify points that the authors misunderstood:

- I did not ask to generate multiple sequence alignments - simply use the guide trees generated by Clustal-W, which only needs pairwise sequence alignments that the authors are generating by other means. This just would remove the need for the authors implement their approach saving them some time. But again, this paper is not about computational method development, their approach would generate same or very similar results.

- "In our method the graph is simply split by introducing the threshold (i.e., 'splitting threshold') and removing the edges lower than the threshold after which subgraphs representing structural variants are identified."  that is exactly what CAST does as well to generate clusters in the of form cliques - and in the case of this manuscript, those clusters simply represent SVs. Application is different, method is the same (again, this is not a methods paper, so it is fine).

I agree with the authors about the issues related to the use of short reads. The analysis performed in this manuscript focuses on ~10 kb downstream of the blaNDM gene (Figure 2), and the median contig length generated by the assembly process is 10 Kb (Supp Fig 1). Also Supp Fig 1

demonstrates that a great majority contigs generated by short read assembly exceed this median length, and authors discarded shorter contigs. Additional PacBio and Ion Torrent data was used to improve assemblies used in the project and several thousands of reference genomes were obtained from Enterobase, Genbank, and RefSeq. Therefore I do not agree that the study is fundamentally flawed within the scope and aims of this project.

Reviewer #4:

Remarks to the Author:

This is an interesting article describing the evolution and dynamics of a drug resistant gene (bla_{NDM}). I found this article well written and with some clever analyses. It has already been reviewed by at least two experts.

The main point about the unreliability of short-read data raised by Reviewer 1 is key and I think the authors have tried to address it to some extent by explaining that they did not only use short-read data. To decide whether this is satisfactory I would think that there needs to be some careful validation of how much short-read data can be tolerated before spurious calls are made, which I do think the authors have tried to address in around line 397. This point is not my strong suit, so I do think it is important that the revisions satisfactorily address concerns raised by the previous Reviewer (#1).

I will focus on the molecular dating, which the other reviewers have not flagged, but which I consider important. These are relatively simple points, but I do consider them essential.

Major points

- Please do not include a p-value for the root-to-tip regression. This is statistically invalid, but unfortunately very common in some high-profile papers. The date randomisation is only for visual inspection and it does not have clear cut-offs for deciding whether there is temporal signal. Arguably one wants a slope that is positive and a 'reasonable' high R^2 , but it is all too arbitrary to refer to this as a test of temporal signal. I suggest presenting the regression and stating that it did not reveal obvious outliers or problems with the data, but not using its statistics as diagnostics.

- Were there 10,000 date randomisations? This is an absolute record number. Please check that this is correct. Personally, I find the date-randomisation test too conservative, and this prefer to use BETS instead (<https://academic.oup.com/mbe/article/37/11/3363/5867920?login=true>), but will not insist on it here because I do not think it is necessary.

- One point that is very important in all Bayesian molecular clock analyses is to state the prior on the tree prior parameters. In particular, I noticed that the constant-coalescent gave a somewhat older t_{mrca} than the other tree priors. This sometimes happens because the population size prior is very vague or with a lot of density concentrated on large values. This is not a problem here, but it is important to state clearly.

- BEAST2 uses a uniform prior from 0 to infinity as on the clock rate by default. I dislike this prior because it is improper, but it often behaves well if the data have good temporal signal. I think it is better to use something like a gamma that is not overly skewed. Again, not that this is a big problem here, but please do state the prior on the clock rate.

- I think there need to be a few lines justifying the choice of tree priors. Arguably the molecular clock model is more important than the tree prior to estimate dates, but I did not see here analyses using other clock models, such as the relaxed ones. If this was the first round of revisions, I would probably insist on this, but adding a few lines explaining why these model combinations were chosen and their assumptions would suffice.

The reason why I raise this point is that many papers simply run an arbitrary set of models without considering their assumptions and then it become standard practice, but with no clear reasoning behind it.

- Finally, I did not see any attempts to mask sites that may be prone to errors. If there is a small number of them, then this is not too problematic here, but if it is a large number it would be helpful to mask them prior to phylogenetic and molecular dating analyses. At the very least I would like to see some text explaining this potential problem.

Response to Reviewers

We wish to thank the editor and the reviewers for their thorough assessment and useful comments.

In our revised manuscript, we have addressed all the reviewers' comments. Accordingly, we include a point-by-point response to reviewers' comments below. Our responses are shown in red immediately under each of the reviewer's comments.

Yours sincerely,

Mislav Acman

(on behalf of all the co-authors)

REVIEWER COMMENTS

Reviewer #1 (Remarks to the Author):

This revision does not seem to have taken on board any of the comments made about the previous submission and the responses to many of those comments do not actually address them. This is a surprising approach. A few references covering what is already known were included. However, it's not clear what the analysis presented adds to what is known already from those references. Hence, it still does not increase what is known about the dissemination of the blaNDM gene.

This manuscript presents a new, systematic, and purely data driven approach for studying mobile genetic elements that aims to by bypass possible biases in our current body of knowledge. As such, it attempts to keep any prior about the blaNDM gene as objective and diffuse as possible and reconstructs the spread of the gene solely from the signal in the sequence dataset. Once more we would like to point out that it is not our intention to diminish any of the previous findings about the dissemination of blaNDM. However, this is not a review paper, and we believe that we reference and include the previously key findings relevant to provide a background to our results.

The dating story (an addition) fails to account for sequence quality, so how many of the SNP are simply errors and how many real is not addressed. This invalidates this aspect of the work.

Thank you for pointing this out. The sequence quality of SRA samples has now been assessed and included in the manuscript. We found no problems with quality of SRA assemblies (lines 89-91). We found >30x coverage for the vast majority of SRA contigs. In 10 contigs, we found two of the SNPs used in the dating analysis had a lower Phred quality score. However, this score is still acceptable (>20) and masking these SNPs would likely not impact the dating results considering these are found on few of the more recent SRA samples (i.e. more recent than 2012). This analysis has been summarized in Supplementary Figure 2 and a new Methods section on quality assessment (lines 356-364). The sequence quality reports from other databases are not available and cannot be derived without raw sequencing reads.

In addition, the methodology is unnecessarily complicated. For example, the contig containing Tn125 can be found simply in draft genomes using BLAST searches.

We do not agree with the reviewer that our methodology is 'complicated'. It is correct that one can simply do a BLAST search to recover contigs carrying Tn125, however, this was not the point of our analysis. Finding structural variants uncovers more information than just identifying sequences bearing Tn125. It provides evidence for ancestral background, shows which structural variations are globally or locally dominant (depending on the analysis), can help provide context for specific genome reshuffling events, helps discover new transposable elements, provides information about the longest alignable region (needed for molecular dating), and many more. Furthermore, our methodology of uncovering structural variants does not require any prior knowledge about the mobile elements. In the case of BLAST-ing for specific elements, one would need to know *a priori* the sequence of, for instance, Tn125. Assuming one does know nothing about the genetic context of a particular resistance element, how would one identify transposons and other structural variations surrounding their gene of interest?

Overall, the findings lag way behind what is already known. Splitting information about the context upstream from the context downstream is a serious concern as the complete picture requires every pair to be linked and other available Methods supply this information.

This is the current limitation of the method, and there is no straightforward solution at this stage. We have been open about this exact limitation in the manuscript (lines 131-134; 397-400).

Overall the text was essentially hearsay as how the findings were reached is not adequately covered. For example, the plasmid type can be identified if the plasmid sequence is complete but generally it will not be possible to deduce even plasmid association from draft genomes. How these issues were dealt with is not

mentioned or explained. How many had a contig containing both the NDM gene and identifiable plasmid sequence?

Plasmid typing is performed by screening contigs against the PlasmidFinder replicon database. A plasmid type is defined based on a small and partially conserved portion of the sequence involved in replication of the plasmid genome. Screening contigs against a database of plasmid sequences helped us identify putative plasmid sequences. These contigs are of similar length and have high sequence similarity as the reference plasmid in the database. The process of identifying putative plasmid sequences and plasmid types is explained in the methods section lines 379-387. The number of contigs containing both the blaNDM gene and identifiable plasmid sequence (i.e. plasmid sequence from the reference database) is 3,599 (line 110).

It is well known that some sequencing technologies are more error prone than others. To look in detail at SNPs, the poor-quality genomes must first be filtered out. Even then, there are problems as conjugative transfer can fix and then amplify a sequence variation.

Any problems in our analysis arising from the lower quality genomes are accounted for by the sheer number of sequences analysed. We demonstrated there are no predominant quality issues with our hybrid or SRA assemblies. Also, majority of analysed sequences come from well-curated RefSeq database.

The text in multiple Figures is too small or of insufficient contrast with the background to be read easily.

Thank you for your concern. We are ready to make any further adjustments to the figures as requested by the editorial team.

Reviewer #2 (Remarks to the Author):

As I have stated in my first review, this is not a novel methods paper, therefore I will not further pursue methodological novelty in my assessment. Just a couple of comments to clarify points that the authors misunderstood:

- I did not ask to generate multiple sequence alignments - simply use the guide trees generated by Clustal-W, which only needs pairwise sequence alignments that the authors are generating by other means. This just would remove the need for the authors implement their approach saving them some time. But again, this paper is not about computational method development, their approach would generate same or very similar results.

Apologies for misunderstanding and thank you for clarification and for your constructive input. We believe there are multiple issues relating to using ClustalW in this regard. Based on our understanding, the guide tree in ClustalW is produced from a distance matrix which represents all the pairwise distances of the sequences. For each pair, the distance is computed using k-tuple distance measure. The guide tree is then produced using fast UPGMA method. If one is comparing k-mers between pairs of sequences, then the distance measure is influenced by both the length of these sequences and the (dis)similarity between individual bases. However, from such score, we cannot distinguish the information about the length of homology between pairs of sequences which we are ultimately interested in.

In our approach, the length of homology is proxied by the length of the pairwise alignment, and it is this score we use to cluster our sequences by using network analysis. One solution would be to supply a custom distance matrix to the ClustalW, but we could equally use an independent implementation of UPGMA to cluster it. The problem with using UPGMA is that it produces an ultrametric tree so the branch distances would not be meaningful, but perhaps other clustering methods such as Neighbour Joining or Hierarchical Clustering could be used instead. However, here arises another problem concerning the distance matrix. The matrix of alignment lengths is not a distance matrix, and one would need to find a meaningful way to convert it into one prior to using the aforementioned algorithms. Such distance matrix should preserve the information about length of homology between sequences so that the trees produced by the algorithms can be easily interpreted. In our analysis, we did consider using various clustering algorithms, but eventually opted for a

network analysis approach. We believe it to be a more intuitive way to look at homology as it allowed us to work directly with the alignment length matrix.

- "In our method the graph is simply split by introducing the threshold (i.e., 'splitting threshold') and removing the edges lower than the threshold after which subgraphs representing structural variants are identified."  that is exactly what CAST does as well to generate clusters in the of form cliques - and in the case of this manuscript, those clusters simply represent SVs. Application is different, method is the same (again, this is not a methods paper, so it is fine).

Thank you for bringing our attention back to this point. We have now included a sentence: *Similar clustering algorithms have been previously described, eg. CAST, but were used for a different application.* (lines 412-413)

I agree with the authors about the issues related to the use of short reads. The analysis performed in this manuscript focuses on ~10 kb downstream of the blaNDM gene (Figure 2), and the median contig length generated by the assembly process is 10 Kb (Supp Fig 1). Also Supp Fig 1 demonstrates that a great majority contigs generated by short read assembly exceed this median length, and authors discarded shorter contigs. Additional PacBio and Ion Torrent data was used to improve assemblies used in the project and several thousands of reference genomes were obtained from Enterobase, Genbank, and RefSeq. Therefore I do not agree that the study is fundamentally flawed within the scope and aims of this project.

We appreciate that you recognize the concerns raised by Reviewer #1 are unfounded in the case of our manuscript.

Reviewer #4 (Remarks to the Author):

This is an interesting article describing the evolution and dynamics of a drug resistant gene (blaNDM). I found this article well written and with some clever analyses. It has already been reviewed by at least two experts.

The main point about the unreliability of short-read data raised by Reviewer 1 is key and I think the authors have tried to address it to some extent by explaining that they did not only use short-read data. To decide whether this is satisfactory I would think that there needs to be some careful validation of how much short-read data can be tolerated before spurious calls are made, which I do think the authors have tried to address in around line 397. This point is not my strong suit, so I do think it is important that the revisions satisfactorily address concerns raised by the previous Reviewer (#1).

We are very glad to hear you appreciate our work. We believe the issues raised by Reviewer #1 regarding short-read data have now been addressed. In our revised manuscript, we have assessed the quality of contigs obtained from SRA assemblies, as well as quality of individual SNPs used in molecular dating. We found both to be sufficiently good and not compromising our results. The quality assessment has been summarized in Supplementary Figure 2 and new Methods section on quality assessment (lines 356-364).

I will focus on the molecular dating, which the other reviewers have not flagged, but which I consider important. These are relatively simple points, but I do consider them essential.

Major points

- Please do not include a p-value for the root-to-tip regression. This is statistically invalid, but unfortunately very common in some high-profile papers. The date randomisation is only for visual inspection and it does not have clear cut-offs for deciding whether there is temporal signal. Arguably one wants a slope that is positive and a 'reasonable' high R², but it is all too arbitrary to refer to this as a test of temporal signal. I suggest presenting the regression and stating that it did not reveal obvious outliers or problems with the data, but not using its statistics as diagnostics.

Thank you for your suggestion. This has now been corrected and the p-values were removed from the text and from Supplementary Figures 13 and 14. The new statement has been added as you suggested (line 449-450, also line 233).

- Were there 10,000 date randomisations? This is an absolute record number. Please check that this is correct. Personally, I find the date-randomisation test too conservative, and this prefer to use BETS instead (<https://academic.oup.com/mbe/article/37/11/3363/5867920?login=true>), but will not insist on it here because I do not think it is necessary.

10,000 date randomisations is correct. We agree this is quite excessive. Unfortunately, it was an omission which kept propagating with the code reused from a previous analysis. This has now been corrected and number of randomisations has been reduced to 1000 (Supplementary Figure 13 and 14). Also, thank you for your suggestion of using BETs method. We will surely try it out in some future analysis.

- One point that is very important in all Bayesian molecular clock analyses is to state the prior on the tree prior parameters. In particular, I noticed that the constant-coalescent gave a somewhat older tmrca than the other tree priors. This sometimes happens because the population size prior is very vague or with a lot of density concentrated on large values. This is not a problem here, but it is important to state clearly.

Your observations are very keen. Indeed, majority of the priors are set quite loosely which has yielded wide estimates for the emergence of Tn125. In case of population size priors, we have used different coalescent population models in BEAST2 with wide prior distributions provided by default by each model. This was done so because it is extremely difficult to set priors on the population size in case of transposons. Transposons are highly mobile genetic elements present in multiple copies on many plasmids which can be found in multiple species and with varying mobility. Such complexity does not easily fit available population growth models and it is our opinion that a more suitable BEAST models should be developed for dating the emergence of mobile genetic elements, but this goes beyond the scope of this manuscript. The BEAST2 xml configuration files with details of each BEAST2 run are available as Supplementary data 3. However, we appreciate these can be hard to read, and thus encouraged by your suggestions we have rewritten the Methods section explaining the BEAST2 implementation (lines 452-462).

- BEAST2 uses a uniform prior from 0 to infinity as on the clock rate by default. I dislike this prior because it is improper, but it often behaves well if the data have good temporal signal. I think it is better to use something like a gamma that is not overly skewed. Again, not that this is a big problem here, but please do state the prior on the clock rate.

Thank you for your suggestion. The clock rate has now been stated in revised Methods section (lines 452-462).

- I think there need to be a few lines justifying the choice of tree priors. Arguably the molecular clock model is more important than the tree prior to estimate dates, but I did not see here analyses using other clock models, such as the relaxed ones. If this was the first round of revisions, I would probably insist on this, but adding a few lines explaining why these model combinations were chosen and their assumptions would suffice.

The reason why I raise this point is that many papers simply run an arbitrary set of models without considering their assumptions and then it become standard practice, but with no clear reasoning behind it.

We have restrained from using relaxed clock models for three reasons: computational costs, recent emergence of these mobile elements, and lack of host structure in phylogenetic trees. Please find the justification for the choice of priors included in our rewritten Methods section (lines 452-462).

- Finally, I did not see any attempts to mask sites that may be prone to errors. If there is a small number of them, then this is not too problematic here, but if it is a large number it would be helpful to mask them prior to phylogenetic and molecular dating analyses. At the very least I would like to see some text explaining this potential problem.

Thank you for your comment. We believe this point has been partially addressed in the quality assessment analysis (lines 356-364) where we found no sites of concerning quality. However, we are unable to assess the quality of other sites used in dating analysis as these originate from pre-assembled sequences from various databases. We manually inspected alignments used in molecular dating and found no concerning sites. This has now been stated in lines 433-435.

Reviewers' Comments:

Reviewer #1:

Remarks to the Author:

The problems with this manuscript are many and fundamental. Various issues that have been raised previously by reviewers have been simply dismissed rather than addressed. Hence they remain. This is not a scientific approach.

The text is very hard to follow and the information supplied is insufficient to know how the conclusions were reached and to assess whether they are correct. Hence the conclusions amount to hearsay.

It appears that the authors are not familiar with mobile elements and a huge number of statements (far too many to list) are simply incorrect.

The methodology is not novel; others have used the same approach and also obtained results that add little to our understanding of the resistance genes are mobilised. Indeed the view of the mobile element and resistance communities is that the outcomes are simply misleading.

Here are just 3 specific comments.

1. It is not appropriate to combine draft and complete genome contigs. At the very least the analysis should be done on each group separately. The authors appear to be unaware of the causes of contig breaks in draft genomes. Were they familiar with this, they would not have included these contigs which will often be quite short especially if IS26 or ISAb125 or various other mobile elements are present in the genome. Hence, the analysis is of extremely questionable value.
2. What matters is the context on the left AND the right for each contig. This is a massive problem that is not addressed.
2. Plasmid associations. First, PlasmidFinder cannot find Acinetobacter plasmids. Second, the presence of plasmid-derived contigs and NDM-containing contigs in the same genome DOES NOT indicate an association. Associations can ONLY be derived from complete plasmid sequences. The text of this section gives no indication as to what exactly was done.
3. With a total of 17 single aa changes in NDM and 17 (or a few more?) base changes to work with, it is unlikely that BEAST or any other programme could estimate the time of origin. Moreover there are effects due to transfer into new hosts fixing SNPs that have not been accounted for.

A minor issue: this reviewer is not aware of any evidence that ISCR, originally called CR, are indeed mobile elements of the IS type. If you have evidence please supply it.

Reviewer #2:

Remarks to the Author:

The authors adequately responded to my comments. I have no further questions.

Reviewer #4:

Remarks to the Author:

The authors have substantially revised their manuscript to address comments by several reviewers and me.

The molecular dating looks more solid to me and much better explained. I particularly appreciate the improved explanation of why models were selected, which I think will be valuable for readers. All my concerns and comments have been satisfactorily addressed.

Response to Reviewers

We wish to thank the editor and the reviewers for their invaluable input during this revision process. Your advice, comments, and suggestions helped to make the manuscript more accessible to readers and our findings more transparent and easier to replicate.

In our revised manuscript, we have addressed all editorial requests as well as the final comments by reviewer #1. As always, we include a point-by-point response to reviewer 1#'s comments below. Our responses are shown in red.

Yours sincerely,

Mislav Acman

(on behalf of all the co-authors)

REVIEWERS' COMMENTS

Reviewer #1:

The problems with this manuscript are many and fundamental. Various issues that have been raised previously by reviewers have been simply dismissed rather than addressed. Hence they remain. This is not a scientific approach. The text is very hard to follow and the information supplied is insufficient to know how the conclusions were reached and to assess whether they are correct. Hence the conclusions amount to hearsay. It appears that the authors are not familiar with mobile elements and a huge number of statements (far too many to list) are simply incorrect. The methodology is not novel; others have used the same approach and also obtained results that add little to our understanding of the resistance genes are mobilised. Indeed the view of the mobile element and resistance communities is that the outcomes are simply misleading.

We believe we engaged with your criticisms of the work in good faith and regret we could not sway your opinion. More positively, we appreciate that the robust and lengthy reviewing process led to a much-improved end product.

Here are just 3 specific comments.

1. It is not appropriate to combine draft and complete genome contigs. At the very least the analysis should be done on each group separately. The authors appear to be unaware of the causes of contig breaks in draft genomes. Were they familiar with this, they would not have included these contigs which will often be quite short especially if IS26 or ISAb125 or various other mobile elements are present in the genome. Hence, the analysis is of extremely questionable value.

The distinction between 'draft' and 'complete' genome contigs is artificial and largely irrelevant in this context. A more meaningful distinction may be between plasmids represented by several contigs, and 'closed' (complete) plasmids (i.e., reconstructed as a single contig). The data we produced ourselves provides a fair number of new 'closed' plasmids. That said, the fact that plasmid sequences could be closed does not always mean its sequence is of higher quality. Due to the extremely high plasticity of plasmid genomes, sequencing reads cannot be mapped to a 'reference plasmid genome'. Instead, contigs are constructed using a *de novo* assembly method. Thus, such contigs are inherently 'draft' even they represent closed plasmids. As such, we cannot see any good reason why contigs of sufficient length should not be combined. There are multiple causes behind contig breaks, and as far as we know, there is currently no universal approach to identify the underlying causes. Also, we would like to remind the reviewer that there is a section in the manuscript discussing possible causes for contig breaks (lines 189-206).

2. What matters is the context on the left AND the right for each contig. This is a massive problem that is not addressed.

We agree with the reviewer: the biggest shortcoming of our approach is the inability to explore both left and right flanks simultaneously. As noted before, this property is inherent to the method and, in our opinion, not easy to overcome. Nevertheless, we believe this does not present a problem in the context of *bla*_{NDM} as the alignment rapidly breaks down upstream of the gene thus making the upstream context much less informative. Furthermore, we do agree readers should be aware of this limitation and hence we included a paragraph in the discussion further expanding on this issue (lines 315-322)

2. Plasmid associations. First, PlasmidFinder cannot find *Acinetobacter* plasmids. Second, the presence of plasmid-derived contigs and NDM-containing contigs in the same genome DOES NOT indicate an association. Associations can ONLY be derived from complete plasmid sequences. The text of this section gives no indication as to what exactly was done.

It is true that PlasmidFinder cannot find *Acinetobacter* plasmids, and one of the main reasons for this is because *Acinetobacter* plasmids are understudied and underrepresented. However, we would like to remind the reviewer that *Acinetobacter* sequences represent only a small portion of our dataset. This point was previously raised by the reviewer, and we have accordingly implemented an additional plasmid identification method where we screened NDM-positive contigs against a database of complete bacterial plasmids (which does contain *Acinetobacter* plasmids). These results have been used in subsequent analysis. More information on plasmid screening can be found in the methods section (lines 381-395)

3. With a total of 17 single aa changes in NDM and 17 (or a few more?) base changes to work with, it's unlikely that BEAST or any other programme could estimate the time of origin. Moreover there are effects due to transfer into new hosts fixing SNP that have not been accounted for.

We used two different Bayesian molecular dating software, several population models, and very loose priors in order to be able to detect and avoid caveats you mention (lines 438-477). All these approaches provided congruent results. In addition, such analysis setup, aided by fewer mutations present in the aligned region, has resulted in a wide estimation of the emergence time. While our estimations are not precise enough to pinpoint the exact year/decade of *bla*_{NDM} mobilisation in its ancestral transposons, they allow us to falsify previous claims for a very recent (5-10 year ago) emergence, and to give a temporal perspective on *bla*_{NDM} emergence, origin and mobility patterns. Furthermore, the effect of different mutation rates between bacterial hosts would be more relevant on a longer temporal scale, i.e., thousands of years. In this case (<100 years), it is unlikely to have a major impact, especially considering the wide confidence intervals.

A minor issue: this reviewer is not aware of any evidence that ISCR, originally called CR, are indeed mobile elements of the IS type. If you have evidence please supply it.

IS Common Region elements (ISCR) were included in the IS91 family (employing HUH-Tnps) because they possess some key motifs common to IS91-like elements. Also, like other IS91 elements, they do not contain IRs and are presumed to employ a rolling-circle form of transposition. More information can be found in the following review articles:

<https://doi.org/10.3103/S0891416812040040>

<https://doi.org/10.1128/MMBR.00048-05>

Reviewer #2 (Remarks to the Author):

The authors adequately responded to my comments. I have no further questions.

We are delighted to hear you find our revisions satisfactory. Thank you for your input in this review process.

Reviewer #4 (Remarks to the Author):

The authors have substantially revised their manuscript to address comments by several reviewers and me.

The molecular dating looks more solid to me and much better explained. I particularly appreciate the improved explanation of why models were selected, which I think will be valuable for readers. All my concerns and comments have been satisfactorily addressed.

We are happy to hear you find our revisions satisfactory. Thank you very much for your comments and suggestions regarding molecular dating analysis.